# An Evolutionary Approach to Dynamic Introduction of Tasks in Large-scale Multitask Learning Systems

## Abstract

Multitask learning assumes that models capable of learning from multiple tasks can achieve better quality and efficiency via knowledge transfer, a key feature of human learning. Though, state of the art ML models rely on high customization for each task and leverage size and data scale rather than scaling the number of tasks. Also, continual learning, that adds the temporal aspect to multitask, is often focused to the study of common pitfalls such as catastrophic forgetting instead of being studied at a large scale as a critical component to build the next generation artificial intelligence. We propose an evolutionary method capable of generating large scale multitask models that support the dynamic addition of new tasks. The generated multitask models are sparsely activated and integrates a task-based routing that guarantees bounded compute cost and fewer added parameters per task as the model expands. The proposed method relies on a knowledge compartmentalization technique to achieve immunity against catastrophic forgetting and other common pitfalls such as gradient interference and negative transfer. We demonstrate empirically that the proposed method can jointly solve and achieve competitive results on 69 public image classification tasks, for example improving the state of the art on a competitive benchmark such as cifar10 by achieving a 15% relative error reduction compared to the best model trained on public data.

## 1 Introduction

The success of machine learning continues to grow as it finds new applications in areas as diverse as language generation (Brown et al., 2020), visual art generation (Ramesh et al., 2021), chip design (Mirhoseini et al., 2020), protein folding (Senior et al., 2020) and competitive sports (Silver et al., 2016; Vinyals et al., 2019). The vast majority of machine learning models are designed and trained for a single task and specific data modality, and are often trained by starting with randomly initialized parameters, or with limited knowledge transfer from a pre-trained model. While this paradigm has shown great success, it uses a large amount of computational resources, and does not leverage knowledge transfer from many related tasks in order to achieve higher performance and efficiency.

The work presented in this paper is based on the intuition that significant advances can be enabled by dynamic, continual learning approaches capable of achieving knowledge transfer across a very large number of tasks. The method described in this paper can dynamically incorporate new tasks into a large running system, can leverage pieces of a sparse multitask ML model to achieve improved quality for new tasks, and can automatically share pieces of the model among related tasks. This method can enhance quality on each task, and also improve efficiency in terms of convergence time, amount of training examples, energy consumption and human engineering effort.

The ML problem framing proposed by this paper can be interpreted as a generalization and synthesis of the standard multitask and continual learning formalization, since an arbitrarily large set of tasks can be solved jointly. But also, over time, the set of tasks can be extended with a continuous stream of new tasks. Furthermore, it lifts the distinction between a pretraining task and a downstream task. As new tasks are incorporated, the system searches for how to combine the knowledge and representations already present in the system with new model capacity in order to achieve high quality

for each new task. Knowledge acquired and representations learned while solving a new task are available for use by any future task or continued learning for existing tasks.

We refer to the proposed method as "mutant multitask network" or $\mu$2Net. This method generates a large scale multitask network that jointly solves multiple tasks to achieve increased quality and efficiency for each. It can continuously expand the model by allowing the dynamic addition of new tasks. The more accumulated knowledge that is embedded into the system via learning on previous tasks, the higher quality the solutions are for subsequent tasks. Furthermore, new tasks can be solved with increasing efficiency in terms of reducing the newly-added parameters per task. The generated multitask model is sparsely activated as it integrates a task-based routing mechanism that guarantees bounded compute cost per task as the model expands. The knowledge learned from each task is compartmentalized in components that can be reused by multiple tasks. As demonstrated through experiments, this compartmentalization technique avoids the common problems of multitask and continual learning models, such as catastrophic forgetting, gradient interference and negative transfer. The exploration of the space of task routes and identification of the subset of prior knowledge most relevant for each task is guided by an evolutionary algorithm designed to dynamically adjust the exploration/exploitation balance without need of manual tuning of meta-parameters. The same evolutionary logic is employed to dynamically tune the hyperparameters multitask model components.

## 2 RELATED WORK

The main novelty of the presented work is to propose and demonstrate a method that jointly provides all of the following properties: 1) ability to continually learn from an unbounded stream of tasks, 2) automate the selection and reuse of prior knowledge and representations learned for previous tasks in the solving of new tasks, 3) search the space of possible model architectures allowing the system to dynamically extend its capacity and structure without requiring random initialization, 4) automatically tune the hyperparameters of both the generated models and the evolutionary method, including the ability to learn schedules for each hyperparameter, rather than just constant values, 5) ability to optimize for any reward function, also including non-differentiable factors, 6) immunity from catastrophic forgetting, negative transfer and gradient interference, 7) ability to extend any pre-existing pre-trained model, including extending its architecture and adapting the domain on which such model have been trained to other domains automatically, 8) introduction of a flexible access control list mechanism that allows expression of a variety of privacy policies, including allowing the use or influence of task-specific data to be constrained to just a single task or to a subset of tasks for which data or higher-level representation use should be permitted. Different lines of research have focused on distinct subsets of the many topics addressed by the proposed method. In this section we highlight a few cornerstone publications. Refer to Appendix A for an extended survey.

Different methods have been proposed to achieve **dynamic architecture extensions** (Chen et al., 2016; Cai et al., 2018), some also focusing on an unbounded stream of tasks (Yoon et al., 2018), or achieving immunity from catastrophic forgetting (Rusu et al., 2016; Li & Hoiem, 2018; Rosenfeld & Tsotsos, 2020). Unlike our work, these techniques rely on static heuristics and patterns to define the the structural extensions, rather than a more open-ended learned search process. **Neural architecture search** (NAS) (Zoph & Le, 2017) methods aim to modularize the architectural components in search spaces whose exploration can be automated with reinforcement learning or evolutionary approaches (Real et al., 2019; Maziarz et al., 2018). More efficient (but structurally constrained) parameter sharing NAS techniques (Pham et al., 2018; Liu et al., 2019a) create a connection with routing methods (Fernando et al., 2017) and **sparse activation** techniques, that enable the decoupling of model size growth from compute cost growth (Shazeer et al., 2017; Du et al., 2021). Evolutionary methods have also been applied with success for **hyperparameter tuning** (Jaderberg et al., 2017).

Cross-task **knowledge transfer** has gained popularity, especially through transfer learning from a model pre-trained on a large amount of data for one or a few general tasks, and then fine-tuned on a small amount of data for a related downstream task. This approach has been shown to be very effective in a wide variety of problems and modalities (Devlin et al., 2019; Dosovitskiy et al., 2021). Large scale models have recently achieved novel transfer capabilities such as few/zero shot learning (Brown et al., 2020). More complex forms of knowledge transfer such as **multitask training** or **continual learning** often lead to interesting problems such as catastrophic forgetting (McCloskey & Cohen, 1989; French, 1999), negative transfer (Rosenstein, 2005; Wang et al., 2019) or gradient interference

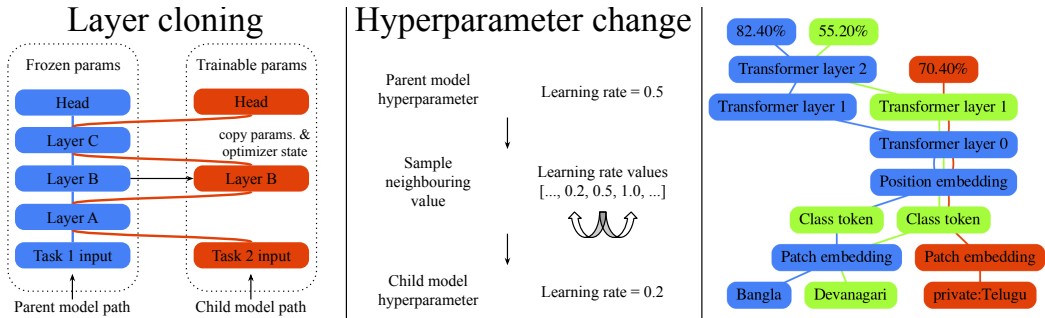

Figure 1: Graphical representation of the two mutation types used by the proposed method: layer cloning mutation (left) and hyperparameter change (center). The graph on the right represents the model generated by the preliminary experiment described in Section 4. The bottom nodes display the task names, the top nodes display the validation accuracy, and internal nodes are represented with the color of the task that has last updated the parameters of the corresponding layer.

(Chen et al., 2018; Yu et al., 2020). Research on these topics mostly focuses on approaches such as weighted combination methods (Liu et al., 2019b; Sun et al., 2020b) or gradient transformations (Sener & Koltun, 2018; Kendall et al., 2018), also methods automating knowledge selection at a layer level was proposed (Sun et al., 2020a).

## 3 EVOLUTIONARY METHOD

This section defines the proposed method capable of generating a dynamic multitask ML system. The multitask system is initialized with one *root model*. This model can be either pretrained or randomly initialized. During the evolutionary process, the proposed method searches for the best model for a single task at a time, referred to as the *active task*. During the active phase of a task, a population of models trained on the active task is evolved: the *active population*. The first time a task becomes active, its active population is empty. For subsequent iterations, the active population is initialized with all the models trained on the active task that have been retained from previous iterations. The active population is iteratively extended by: 1) sampling a *parent model* (Section 3.1), 2) applying to the parent model a sampled set of mutations (Section 3.2) to produce a *child model*, 3) performing cycles of training and validation in order to train and score the child model. Each trained model is assigned a *score* (Section 3.3). Early population pruning is performed by discarding the models that did not achieve a better score then their parent. An active phase is composed of multiple *generations* in which multiple batches of child models are sampled and trained in parallel. At the end of a task active phase, only its best scoring model is retained as part of the multitask system. A task can become active multiple times. Details of the method are reported below (and in Algorithm 1).

### 3.1 PARENT MODEL SAMPLING

The first attempt to sample a parent model for the active task is done over the active population of models for that task. The models in the active population are visited in decreasing order of score, starting with the highest scoring one. Each model, $m$, can be accepted as parent with probability: $p_{parent}(m|t) = 0.5^{\#selections(m,t)}$. Where $\#selections(m,t)$ denotes the number of times the candidate model, $m$, has been previously selected as parent to generate a child models for task $t$. If the current candidate parent is not selected, then iteratively the model with the next best score is considered to be selected as parent with probability $p_{parent}(\cdot|t)$. If a full iteration on the active population is completed without a successful parent model selection, then the same method is applied to the randomly sorted list of all remaining models: all the models currently part of the multitask system that were trained on a task different from the current active task, $t$. This fallback list is randomly sorted since these models have not been scored for $t$. As a final fallback a parent is uniformly sampled among all the models currently in the system. This method prioritizes the exploitation of high scoring models that had few attempts at generating an improved model for the active task. But also, in combination with early pruning, it automatically transitions toward a more exploratory behavior in case the higher scoring models are unable to generate an improvement.

### 3.2 Mutations

In this work we consider Deep Neural Networks (DNN) models. DNN are commonly defined by their architecture and hyperparameters. Architectures are composed of a sequence of neural network layers, each mapping an input vector into an output vector of variable dimensions. Hyperparameters specify the configuration details such as the optimizer or data preprocessing configurations. The presented method allows for two types of mutations (Figure 1):

**Hyperparameter mutations** can be applied to modify the configuration inherited from the parent. Each hyperparameter is associated with a sorted list of valid values (Table 1). If a hyperparameter is selected for mutation, then its new value is selected at random among the two neighbouring values in the sorted list (Figure 1). This constraints hyperparameter mutations to only incremental changes. Notice that, every hyperparameter of a child model is set to a single value. However, considering that a child model continues its ancestors training with mutated hyperparameters, then the method can be regarded as capable of defining a piece-wise constant schedule over time for each hyperparameter.

**Layer cloning mutations** create a copy of any parent model layer that can be trained by the child model. If a layer of the parent model is not selected for cloning, then it is shared with the child model in a frozen state to guarantee immutability of pre-existing models. Child models can train only the cloned copies of the parent layers. The cloned layers are trained with a possibly modified version of the parent optimizer. The configuration of the child optimizer is defined by the mutated hyperparameters. If such optimizer is of a type that stores a state (i.e. momentum), then the state is also cloned from the state saved by the ancestor that has last trained the cloned layer. Notice that, a trainable layer may be followed by frozen layers. In this case the gradients for the trainable layer are propagated through the frozen layers and applied only to the parameters of the trainable layers while frozen parameters are left unchanged. The head layer is always cloned since it always needs to be trainable. If a child model is trained on a task different from the parent's task, then a new randomly initialized head layer is created with output shape matching the number of classes of the new task.

Each possible layer cloning or hyperparameter mutation is independently sampled to be applied with probability $\mu$. $\mu$ is itself a hyperparameter that is mutated by the evolutionary process. Thus demonstrating that automatic tuning is not only applied to selecting the hyperparameters of the generated models, but can also be applied to self-tune the configuration of the evolutionary algorithm.

### 3.3 Training and scoring

A newly sampled child model is trained on the active task for a given number of epochs. The model is evaluated on the validation set after each epoch. At each intermediate evaluation, the child model is assigned a score that the evolutionary algorithm aims to maximize. The score can be defined to optimize a mixture of factors such as validation quality, inference latency, training compute or model size, depending on the applications requirements. The presented experiments aim to compare against the state of the art for a large number of tasks without any size or compute constraint. Therefore, the validation accuracy is used directly as the score without additional factors. After training, only the parameters and optimizer state of the version of the child model achieving best score are retained.

### 3.4 Discussion and properties

Notice that, none of the defined mutation actions or the evolutionary algorithm allow the creation of child models that can alter the parent model in any way. Once a model has been trained, the parameters storing its knowledge cannot be modified. This method guarantees immunity from **catastrophic forgetting**, since the knowledge of a trained model is always preserved. It also provides a solution to **negative transfer**, since it automates the selection of the knowledge that is most relevant for each new task. Furthermore, it also avoids **gradient interference**, that can arise when multiple gradients are synchronously applied to the same set of parameters. Nonetheless, models for new tasks can use knowledge and representations from prior tasks and even extend these to improve or specialize them.

The method compartmentalizes the knowledge of each task in a subset of components, allowing the implementation of different **dataset privacy control** policies. For example, we can introduce private tasks that can benefit from all the public knowledge embedded in the multitask system but are able to withhold the knowledge and representations derived from their private dataset from being used by other tasks. This is achieved by preventing other tasks from using or cloning components trained on

Table 1: Hyperparameters valid values. Bold vales are defaults. This search space consists of a parametrization of the configuration for the published ViT model definition library.

| |
|---|
| $\mu \in$ [0.10, 0.12, 0.14, 0.16, 0.18, **0.20**, 0.22, 0.24, 0.26, 0.28, 0.30] |
| Learning rate $\in$ [0.0001, 0.0002, 0.0005, 0.001, 0.002, 0.005, **0.01**, 0.02, 0.05, 0.1, 0.2, 0.5] |
| Cosine learning rate schedule warm up ratio $\in$ [0.01, 0.02, 0.05, **0.1**, 0.2, 0.3, 0.4] |
| Momentum $\in$ [0.5, 0.6, 0.7, 0.8, 0.85, **0.9**, 0.95, 0.98, 0.99] |
| Nesterov update $\in$ [**False**, True] |
| Crop input image $\in$ [False, **True**] |
| Cropped area range min $\in$ [0.01, 0.02, **0.05**, 0.1, 0.2, 0.5, 1.0] |
| Cropped aspect ratio range min $\in$ [0.25, 0.5, **0.75**, 1.0] |
| Flip left/right $\in$ [False, **True**] |
| Brightness delta $\in$ [**0.0**, 0.01, 0.02, 0.05, 0.1, 0.2] |
| Contrast delta $\in$ [**0.0**, 0.01, 0.02, 0.05, 0.1, 0.2] |
| Saturation delta $\in$ [**0.0**, 0.01, 0.02, 0.05, 0.1, 0.2] |
| Hue delta $\in$ [**0.0**, 0.01, 0.02, 0.05, 0.1, 0.2] |

private data. This also allows to remove the knowledge learned from the private dataset at any future date by simply removing its components. This private/public distinction can be generalized into an access- control-list mechanism. For example, a set of private tasks can share representations that are withheld from any other task. Privacy control capabilities are empirically demonstrated in Section 4.

### 3.5 EXPERIMENTAL SET UP

This section details the instantiation of the proposed method employed in the experimental analysis. The task type for the presented set of experiment is image classification. This choice allows us to define a large benchmark of publicly available datasets with standardized framing. It also allows us to build on top of state-of-the-art models whose architecture definition and checkpoints are public: the Visual Transformer (ViT) is used as root model (Dosovitskiy et al., 2021).

**Architecture**  Layer cloning mutations can create a copy of any of ViT's layers: 1) *Patch embedding*: the first layer of the model maps the input image into a sequence of embedded tokens, each corresponding to a patch of the input image. 2) *Class token*: a classification token is prepended to the sequence. The final hidden state corresponding to this token is used as the aggregate sequence representation for classification tasks (Devlin et al., 2019). 3) *Position embedding*: the sequence representation is then augmented with an embedding that carries each patch positional information. 4) *Transformer layers*: the sequence representation generated by the input layers is iteratively transformed by a stack of transformer layers (Vaswani et al., 2017). 5) *Model head*: a final fully connected layer mapping the representation produced by the top-most transformer layer into the logits.

**Parameters**  The parameters of the root model can be either randomly initialized or loaded from a checkpoint. The preliminary experiment demonstrates the evolution from random initialization (see Section 4), while the large scale experiment starts from a pretrained large ViT model (see Section 5).

**Hyperparameters**  As default hyperparameters we use those resulting from the extensive study conducted by Steiner et al. (2021): SGD momentum optimizer, cosine decay schedule, no weight decay, gradient clipping at global norm 1 and 386×386 image resolution. The evolutionary method can change the hyperparameters of optimizer, image preprocessing and architecture (see Table 1).

## 4  PRELIMINARY EXPERIMENT

This section describes a small scale preliminary experiment that introduces details of the method application and illustrates the privacy control and initialization capabilities. This experiment demonstrates the ability to generate a multitask model from random initialization and a minimal architecture rather than evolving a pretrained state-of-the-art model. Therefore, a randomly initialized ViT Ti/16 architecture (Steiner et al., 2021) stripped of transformer layers is used as root model. To allow the method to build a capable architecture, we add an extra mutation action that can insert a new randomly initialized transformer layer just before the head layer.

Furthermore, the dataset privacy control technique (see Section 3) is demonstrated by adding a private task. Three numerals recognition tasks are used as benchmark: {bangla, devanagari, telugu}. Telugu is introduced as a private task, so that no other task can access the knowledge introduced into the system by its dataset. However, its models can leverage the knowledge provided by the public tasks.

This short experiment is configured to perform 2 active task iterations for each task. During each active task iteration 4 model generations are produced. In each generation 8 child models are sampled and trained in parallel on each of the 8 TPUv3 cores. The choice of small datasets, architecture and training budget is intended to facilitate reproducibility and fast experimental iterations. The experiment can be reproduced by using the published artifacts and completes in less than 5 minutes.

Figure 1 (right) displays the resulting multitask model solving jointly the 3 tasks. We observe a high degree of cross-task knowledge and components sharing throughout the evolutionary process. Even though the root model has no transformer layers, multiple randomly initialized transformer layers are inserted and trained improving the score of each task. Note that, at any point during the evolution, the components trained on the private task (red) are only used by the private task.

## 5 LARGE SCALE CONTINUAL LEARNING EXPERIMENT

This section reports a single large scale continual learning experiment producing a multitask system jointly solving 69 visual tasks. A pretrained ViT L/16 is used as root model, which has been selected for its pretraining validation accuracy on the imagenet-21k dataset following Steiner et al. (2021).

### 5.1 ViT BENCHMARK

The first tasks introduced to the system are 3 tasks on which ViT was evaluated in Dosovitskiy et al. (2021). This experiment is configured to perform 5 active iterations for each task, and 4 model generations for each iteration. During each generation, 8 child models are trained in parallel on each of the 8 TPUv3 cores. Each model training performs 4 validation cycles. The number of train samples between validation cycles is set to $min(1\ epoch, 128116\ samples)$ to smooth the distribution of compute across datasets of different size. $128116$ is equivalent to $1/10^{th}$ of an epoch of the imagenet2012 training set. The same configuration is applied in following experiments.

This configuration results in 8 epochs for imagenet, and 80 epochs for cifar. This is roughly equivalent to the fine-tuning setup of the baseline model we compare against (Dosovitskiy et al., 2021): 8 epochs for imagenet and 102.4 for cifar. The proposed method can be considered cheaper since: 1) ViT fine-tuning has been repeated multiple times for the hyperparameters tuning process and 2) setting $\mu = 0.2$ results in cheaper training steps, since parameters updates can be skipped for frozen layers and gradient propagation can be skipped for frozen layers at the base of the model (preliminary experiments have shown a 2.5-4% training speed-up attributable to this). In order to provide a fair comparison, as a root model is used the same ViT L/16 architecture, same checkpoint pretrained on the i12k dataset, same $384{\times}384$ image resolution, and optimizer and prepossessing configuration.

Table 2 reports the top 1 test accuracy achieved. $\mu$2Net outperforms fine-tuning with comparable training steps per task. Extending the training with 5 additional tasks iterations leads to moderate gains on imagenet2012 and cifar10. Notice that, for cifar100 the accuracy decreases. This can happen since the best models are selected according the validation accuracy and, as the model gets close to convergence, a small validation accuracy gain may lead to a noisy perturbation of the test accuracy.

To quantify knowledge transfer, we consider the model produced for each task and examine the dataset on which each layer's ancestors were trained. On average, the layers composing the model generated

Table 2: Test accuracy achieved by $\mu$2Net and by fine-tuning a comparable pretrained ViT model.

| Model | imagenet2012 | cifar100 | cifar10 |
|---|---|---|---|
| ViT L/16 fine-tuning (Dosovitskiy et al., 2021) | 85.30 | 93.25 | 99.15 |
| $\mu$2Net after 5 task iterations | 86.38 | 94.75 | 99.35 |
| $\mu$2Net after 10 task iterations | 86.66 | 94.67 | 99.38 |
| $\mu$2Net cont. after adding VTAB-full tasks | **86.74** | 94.67 | 99.41 |
| $\mu$2Net cont. after adding VDD tasks | **86.74** | 94.74 | 99.43 |
| $\mu$2Net cont. after adding all 69 tasks | **86.74** | **94.95** | **99.49** |

Table 3: Test accuracy achieved on the VTAB-full benchmark by: 1) fine-tuning with matching architecture and checkpoint (ViT L/16 i21k) reported by Steiner et al. (2021), 2) the Sup-rotation method (Gidaris et al., 2018) that achieved the best result in the VTAB-full leaderboard (Zhai et al., 2019b), 3-4) $\mu$2Net results after 2 task iterations, 5) and after an additional iteration performed after the VDD benchmark introduction. Underlined models transfer knowledge from VDD tasks.

| Model | caltech101 | cifar100 | dtd | flowers102 | pets | sun397 | svhn | camelyon | eurosat | resisc45 | retinopathy | clevr-count | clevr-dist | dmlab | dspr-loc | dspr-orient | kitti-dist | snorb-azim | snorb-elev |
|---|---|---|---|---|---|---|---|---|---|---|---|---|---|---|---|---|---|---|---|
| Steiner et al. (2021) | **95.5** | 94.1 | 80.3 | **99.6** | 95.0 | 83.4 | 97.4 | 86.4 | 99.0 | 96.6 | 83.3 | 99.8 | 91.7 | 75.6 | 100 | 90.4 | **84.7** | 27.5 | 76.5 |
| Zhai et al. (2019b) | 94.6 | 84.8 | 75.9 | 94.7 | 91.5 | 70.2 | 97.0 | 85.9 | 98.8 | 94.9 | 79.5 | 99.8 | 92.5 | 76.5 | 100 | **96.5** | 82.3 | **100** | **98.4** |
| $\mu$2Net cont. 1st iter. | 92.6 | 94.6 | 79.8 | **99.6** | 95.3 | 84.5 | 97.3 | 87.5 | 99.1 | 96.3 | 83.7 | 99.8 | 93.2 | **76.9** | 100 | 96.2 | 83.0 | 32.3 | 94.5 |
| $\mu$2Net cont. 2nd iter. | 92.6 | 94.6 | 80.5 | **99.6** | 95.3 | 84.8 | 97.8 | 88.4 | **99.2** | **97.0** | **84.0** | 99.8 | **94.0** | **76.9** | 100 | 96.4 | 83.0 | 33.3 | 95.1 |
| $\mu$2Net cont. after VDD | 93.0 | 94.7 | 81.0 | **99.6** | 95.3 | 84.8 | 97.8 | **91.1** | 99.1 | **97.0** | **84.0** | 99.8 | **94.0** | **76.9** | 100 | 96.4 | 82.3 | 33.3 | 95.1 |

for the imagenet2012 task have performed only 60.6% of the training steps on the imagenet2012 dataset, and have received 31.5% of the gradient updates from cifar100 and 7.9% from cifar10. The layers comprising the cifar100 model have performed 42.3% of their training on imagenet2012 and 20.6% on cifar10. And layers comprising the cifar10 model performed 46.1% of training on imagenet2012 and 35.9% on cifar100. The tasks heterogeneity improves the representations produced by the different layers, and results in generally higher performance, as shown in Table 2.

The following sections 5.2 and 5.3 describe the extensions of the system performed by introducing additional benchmarks. After the introduction of each benchmark, we perform an additional iteration on imagenet and cifar tasks to analyze effects of further knowledge enrichment. As a representative example: the VDD benchmark (Section 5.3) includes a low resolution version of cifar100. The model that will be generated for vdd/cifar100 will be a mutation of the current full resolution cifar100 model. Afterward, the additional active task iteration on cifar100 will be performed, and the resulting improved cifar100 model will be a mutation of the low resolution vdd/cifar100 model.

After a final iteration, we note that 99.49 is the best cifar10 accuracy reported for a model trained only on public data: to the best of our knowledge, this constitutes a 15% relative error reduction compared to the 99.40 state of the art achieved by Touvron et al. (2021). Dosovitskiy et al. (2021) achieves 99.50 with a double size ViT-Huge model trained on proprietary data. The achieved cifar100 accuracy is currently outperformed only by Ridnik et al. (2021) (95.10) and Foret et al. (2021) (96.08).

## 5.2 VTAB-full benchmark

Next, we introduce to the system the 19 VTAB-full tasks (Zhai et al., 2019a), plus 5 additional task variants that are not included in the standard evaluation set (Table 8). From this experiment onward, the infrastructure is scaled from 8 to 32 cores, as detailed in Appendix B. The number of task iterations is reduced from 10 to 2. These changes lead to a roughly similar exploratory and training budget per task. However, the increased parallelism results in faster task iterations.

Table 3 reports the achieved results along with reference models that use limited knowledge transfer capabilities. Steiner et al. (2021) reports the quality achieved by fine-tuning a model equivalent to our root model. This outperforms $\mu$2Net on only 2 tasks, even if it has been trained multiple times to perform hyperparameter tuning. Zhai et al. (2019b) reports the results of the best model identified with a large scale study. This state of the art model outperforms $\mu$2Net on 4 tasks. Again, increasing number of task iterations and additional knowledge (VDD) in the system, seem to yield better quality.

Table 4: Test accuracy mean and std.dev. achieved on the VDD benchmark by 3 system replicas.

| Model: $\mu$2Net cont. | imagenet | svhn | cifar100 | gtsrb | daimler | omniglot | ucf101 | aircraft | dtd | flowers |
|---|---|---|---|---|---|---|---|---|---|---|
| 1st iter. | 89.1±0.19 | 98.2±0.23 | 97.2±0.34 | 99.9±0.04 | 99.9±0.03 | 84.5±1.18 | 83.6±1.79 | 64.8±3.57 | 75.2±1.20 | 99.0±0.11 |
| 2nd iter. | 89.2±0.12 | 98.4±0.23 | 97.2±0.40 | 99.9±0.05 | 99.9±0.04 | 84.3±0.27 | 85.7±1.36 | 65.4±3.03 | 76.0±0.49 | 99.2±0.11 |

Table 5: Test accuracy achieved on the Multitask Character Classification Benchmark by $\mu$2Net continued extension with 2 active task iterations and by the model that has set the state of the art using comparable data splits: Jeevan & Sethi (2022) for digits, Kabir et al. (2020) for letters, Ajayan & James (2021) for kmnist, An et al. (2020) for mnist, Hazra et al. (2021) for cmaterdb/bangla. Underlined models reuse knowledge introduced by other character classification tasks. Datasets are listed in decreasing size from the biggest emnist/digits (240k samples) to the smallest telugu (2.5k).

| Model | emnist/ digits | emnist/ letters | kmnist | mnist | omniglot | cmaterdb/ bangla | cmaterdb/ devanagari | cmaterdb/ telugu |
|---|---|---|---|---|---|---|---|---|
| State of the art | 99.77 | **95.88** | 95.40 | **99.91** | – | 99.00 | – | – |
| $\mu$2Net cont. 1st iteration | **99.82** | 93.60 | **98.68** | 99.75 | 98.72 | 98.60 | 96.60 | 97.80 |
| $\mu$2Net cont. 2nd iteration | **99.82** | 93.68 | 98.60 | 99.69 | **99.84** | 99.10 | 98.00 | 99.40 |

## 5.3 VISUAL DOMAIN DECATHLON (VDD) BENCHMARK

The VDD benchmark (Bilen et al., 2017) is introduced next. The ML methodology proposed in this paper, can achieve higher efficiency by focusing the available compute on the training of a single multitask system. Though, the standard approach to measure variance relies on experiment repetitions. This section demonstrates how variance can be measured for any chosen segment of the training. In practice, 2 task iterations are performed to introduce the VDD tasks starting from the state achieved after the introduction of the last benchmark as usual. But, this experiment is run on 3 parallel copies of the system, allowing us to compute variance of the metrics for this set of task iterations.

The VDD benchmark is composed of 10 diverse tasks. This is also reflected in the diverse variance ranges measured (see Table 4). Variance is low for most of the tasks. However, for ucf101 and aircraft is significantly higher. The metrics that have highest correlation with standard deviation are error rate (linear proportionality in log scale) and number of training samples per class (inverse proportionality in log scale) (Figure 4). These can be considered metrics indicative of the complexity of the task. Furthermore, variance decreases with the second iteration: average standard deviation of 0.87 after 1 iteration and 0.61 after the second. These findings can support the intuitive hypothesis that tasks with higher complexity may benefit from more iterations to decrease variance and approach convergence. The next system extension continues from the state of one randomly selected replica.

## 5.4 MULTITASK CHARACTER CLASSIFICATION BENCHMARK

We continue extending the system by adding a set of 8 character classification tasks. Thus offering the opportunity to study knowledge transfer across tasks with high domain correlation.

Table 5 reports the test accuracy achieved with 2 active tasks iterations. We observe that tasks with more training data (left) achieve convergence in the first iteration, this hypothesis is supported by lack of significant accuracy gains with the second iteration. While tasks with less training data (right) show a significant gain from a second training iteration. Smaller tasks use transferred in domain knowledge: bangla top model reuses components that embed knowledge introduced by emnist/letters,

Table 6: Test accuracy achieved on the VTAB-1k benchmark by: 1) fine-tuning ViT L/16 i21k (matching root model) Dosovitskiy et al. (2021), 2) the Sup-rotation method (Gidaris et al., 2018) that achieved the best result in the VTAB-1k leaderboard (Zhai et al., 2019b). Underlined models have at least one ancestor trained on the corresponding full form task. Doubly underlined model inherit directly from the current best model for the matching full form task.

| Model | caltech101 | cifar100 | dtd | flowers102 | pets | sun397 | svhn | camelyon | eurosat | resisc45 | retinopathy | clevr-count | clevr-dist | dmlab | dspr-loc | dspr-orient | kitti-dist | snorb-azim | snorb-elev |
|---|---|---|---|---|---|---|---|---|---|---|---|---|---|---|---|---|---|---|---|
| Dosovitskiy et al. (2021) | 90.8 | 84.1 | 74.1 | 99.3 | 92.7 | 61.0 | 80.9 | 82.5 | 95.6 | 85.2 | 75.3 | 70.3 | 56.1 | 41.9 | 74.7 | 64.9 | 79.9 | 30.5 | 41.7 |
| Zhai et al. (2019b) | 91.7 | 53.7 | 69.5 | 90.8 | 88.1 | 32.8 | 88.5 | 83.4 | 96.0 | 82.0 | 71.1 | 47.3 | 57.2 | 36.6 | 88.3 | 52.1 | 77.1 | 51.6 | 33.7 |
| $\mu$2Net cont. 1st iter. | 87.1 | 89.4 | 77.6 | 99.2 | 94.5 | 57.6 | 97.5 | 86.0 | 98.6 | 93.4 | 78.0 | 91.2 | 59.9 | 47.6 | 58.4 | 96.2 | 81.9 | 32.1 | 92.5 |
| $\mu$2Net cont. 2nd iter. | 89.9 | 90.6 | 78.1 | 99.7 | 94.5 | 57.6 | 97.5 | 86.0 | 98.3 | 93.4 | 83.5 | 99.8 | 90.6 | 76.3 | 100 | 96.2 | 81.7 | 33.7 | 92.5 |

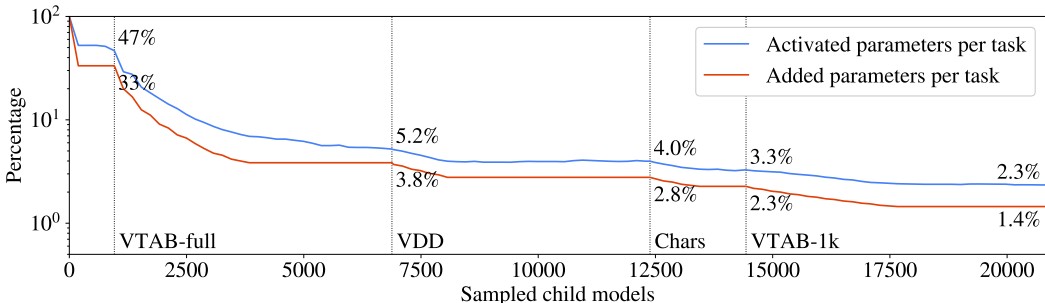

Figure 2: Activated and added parameters per task as percentage with respect to the total number of parameters of the multitask system along the duration the large scale experiment (see Section 5. Vertical lines highlight the start of the introduction for each of the considered benchmark.

while devanagari transfers from omniglot and bangla, and telugu transfers from bangla and devanagari. Furthermore, the achieved quality is comparable to the state of the art published for each task.

### 5.5 VTAB-1K BENCHMARK

The system is further extended by adding the 1k-samples version of the VTAB tasks. Since the system contains already the knowledge learned from the full version of each task, this set allows to study how effective is the proposed method at retrieving knowledge that is already embedded in the system.

Table 6 reports results along with reference models that use limited knowledge transfer capabilities. During the first iteration, the models generated for the short form tasks can retrieve the knowledge of the corresponding full form task also without directly mutating its model, but rather mutating a model having at least one ancestor trained on the full task. For example, the model generated for $flowers102_{1k}$ mutates the $dtd_{1k}$ model, that has 28 ancestors, of which only the $21^{st}$ was trained on $oxford\_flowers102_{full}$. However, that is enough for $flowers102_{1k}$ to achieve 99.2% test accuracy.

After only one task iteration, 5 tasks achieve better test accuracy than the reference models without reusing any knowledge introduced by the corresponding full form task. Particularly interesting is the case of $kitty\text{-}dist_{1k}$ (a.k.a $kitty/closest\_vehicle\_distance_{1k}$), that achieves a strong performance without transferring from the matching full form task but composing the knowledge of related tasks: $kitty/closest\_object\_distance_{full}$ (8 ancestors), $kitty/count\_vehicles_{full}$ (3 ancestors) and $kitty/count\_vehicles_{1k}$ (3 ancestors). Thus learning to estimate distance of the closest vehicle by combining the knowledge of recognizing vehicles and estimating the distance of the closest object. Also, $clevr\text{-}dist_{1k}$ achieves a strong performance by inheriting from the semantically equivalent task $kitti/closest\_object\_distance_{full}$ without reusing the knowledge introduced by $clevr\text{-}dist_{full}$.

## 6 CONCLUSION

We introduced the $\mu$2Net method, aimed at achieving state-of-the-art quality on a large task set, with the ability to dynamically introduce new tasks into the running system. The more tasks are learned the more knowledge is embedded in the system. A ViT-L architecture (307M parameters) was evolved into a multitask system with 13'087M parameters jointly solving 69 tasks. However, as the system grows, the sparsity in parameter activation keeps the amount of compute and the memory usage per task constant. The average added parameters per task decreases by 38% through the experiment, and the resulting multitask system activates only 2.3% of the total parameters per task (see Figure 2 and Table 7). The proposed method allows decoupling the growth of useful representations for solving new tasks from the growth of parameters/compute per task. Furthermore, experimenting with a large number of tasks allowed us to identify different patterns of positive knowledge transfer and composition, achieving higher efficacy on small datasets and across related tasks. The proposed approach to mutations allows to achieve immunity against common pitfalls of multitask systems such as catastrophic forgetting, negative transfer and gradient interference, and demonstrates the key data privacy properties we want to achieve in a continual learning system. Future work can continue to build toward systems that can acquire further capabilities and knowledge across multiple modalities.

All the experiments reported in this paper can be **reproduced** by using the public datasets and published $\mu$2Net code, and can be extended by using the published checkpoints (see Appendix B).

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

## A  EXTENDED RELATED WORK SURVEY

The proposed method is designed to learn a unbounded number of tasks in a continual learning fashion. In such contexts it aims to learn each task with higher quality and efficiency by automating and optimizing the knowledge transfer among any subset of tasks that can provide useful knowledge to one another. The proposed model is designed to be immune from common multitask learning pitfalls: catastrophic forgetting, gradients interference, negative transfer. Cross-task **transfer-learning** has gained popularity, especially through transfer learning from a model pre-trained on a large amount of data for one or a few general tasks, and then fine-tuned on a small amount of data for a related downstream task. This approach has been shown to be very effective in a wide variety of problems across many modalities, including language (Devlin et al., 2019; Raffel et al., 2020) and vision (Dosovitskiy et al., 2021; He et al., 2016).. The success of transfer-learning applications hinges on adequate prior knowledge selection to avoid typical **negative transfer** pitfalls (Rosenstein, 2005; Wang et al., 2019). Common solutions rely on data or model selection techniques, often putting emphasis on the efficiency of the exploration (Zhang et al., 2020; Mensink et al., 2021), also method aiming to automate knowledge selection at a layer level have been proposed Sun et al. (2020a). Transfer learning capabilities are critical for **multitask models**. ML models trained jointly on multiple tasks can be affected by **gradients interference** if any subset of parameters receive gradients jointly from multiple sources (Chen et al., 2018; Yu et al., 2020), and by **catastrophic forgetting** of prior knowledge as new tasks are learned (McCloskey & Cohen, 1989; French, 1999). These knowledge loss problems can be alleviated with weighted combination of tasks (Liu et al., 2019b; Sun et al., 2020b) and gradient transformation methods (Chen et al., 2018; Sener & Koltun, 2018; Kendall et al., 2018). Stronger guarantees are provided by methods that compartmentalize task specific knowledge in dedicated parameter subsets (Rebuffi et al., 2017; Houlsby et al., 2019; Rusu et al., 2016; Rosenfeld & Tsotsos, 2020). Addressing catastrophic forgetting and identifying what subset of parameters/knowledge that is beneficial to share with each ask is also critical for **continual learning** or life long learning methods (McCloskey & Cohen, 1989; French, 1999; Ramesh & Chaudhari, 2022).

The proposed method relies on an evolutionary approach to jointly search the spaces of models architectures, hyperparameters, and prior knowledge selection while optimizing for an possibly multi-factor non-differetiable reward function. The automation of **hyperparameter tuning** has been commonly addressed with Bayesian optimization (Srinivas et al., 2010; Bergstra et al., 2011; Snoek et al., 2012), evolutionary methods have also been explored for this purpose (Jaderberg et al., 2017; Zhang et al., 2011). Hyperparameters tuning can be considered related to the **neural architecture search** (NAS), as architectures can be defined by the selection of a sequence of architectural hyperparameters. Initially, NAS methods have been based on reinforcement learning techniques (Zoph & Le, 2017) but also sample efficient evolutionary approaches have also also proposed (Real et al., 2019; Maziarz et al., 2018). Alternative NAS methods focusing on more efficient parameter-sharing (Pham et al., 2018; Liu et al., 2019a; Kokiopoulou et al., 2019) or optimization for multi-factor quality/cost trade-offs (Tan et al., 2019) have also been explored.

The proposed method is capable to dynamically extend the system, adding capacity or novel structures in an unconstrained fashion. A few methods have been proposed to achieve **dynamic architecture extensions** (Chen et al., 2016; Cai et al., 2018), some also focusing on an unbounded stream of tasks (Yoon et al., 2018; Yao et al., 2020), or achieving immunity from catastrophic forgetting (Rusu et al., 2016; Li & Hoiem, 2018; Li et al., 2019; Rosenfeld & Tsotsos, 2020).

The proposed method is sparsely activated, thus the unbounded growth of knowledge and parameters is decoupled from the growth of computational cost. The growth in capabilities of state of the art models often requires growth in terms of trainable parameters (Kaplan et al., 2020). **Sparse activation** techniques at sub-layer level (Shazeer et al., 2017; Du et al., 2021) or network route level (Fernando et al., 2017) allow to decouple model size growth from compute cost. This is achieved by integrating a **routing technique** that selects the appropriate subset of parameters storing the most relevant knowledge for each task, sample or token/patch.

The ability of jointly solve a **large amount of tasks** is is commonly associated with progress toward Artificial General Intelligence (AGI). Advancements in scaling language models (Brown et al., 2020; Thoppilan et al., 2022) allowed to achieve novel discourse, reasoning and zero/few shot learning capabilities that can be applied to new tasks without/minimal additional training. Recent work aims

to extend these achievements beyond text modality by defining static architectures for an extended subset of modalities (Alayrac et al., 2022; Reed et al., 2022). These are few examples of the ML models contributing to the line of research achieving incremental milestone toward AGI. Though, each models is trained from scratch with considerable resources consumption. The introduction of abstractions allowing to modularize, dynamically extend and reuse these large models may contribute to accelerate the rate of innovation.

## B  EXPERIMENTS REPRODUCIBILITY AND DETAILS

All the experiments reported in this paper can be reproduced by using the following public resources:

- The ViT model definition and checkpoints published by Steiner et al. (2021). These resources are available at https://github.com/google-research/vision_transformer and distributed under the Apache License 2.0.
- Published code of the proposed method: https://... (Anonymized, refer to supplementary material).
- All the used datasets are publicly available via the Tensorflow Datasets image classification, catalog. Refer to https://www.tensorflow.org/datasets/catalog/overview for detailed information regarding each dataset licence and other metadata. Table 8 reports exact dataset splits and reference foor each task.

We also publish the $\mu$2Net checkpoint resulting from the large-scale multitask experiment reported in Section 5. This checkpoint can be used for inference on any of the 69 learned image classification tasks, or for further analysis, or even to be extended with additional tasks or methods. For information about the checkpoint and its license refer to: https://... (Anonymized)

References for mentioned state of the art public model for different tasks are sourced from https://paperswithcode.com as of September 2022.

The VTAB-1k results are reported for reference and are not directly comparable to the state of the art, as the benchmark definition specifies that VTAB-full tasks cannot be used for pre-training.

The initial experiments reported in Sections 4 and 5.1 have been executed on a TPUv3 (Jouppi et al., 2017) machine with 8 cores. While, all the following experiments have been executed on a larger scale infrastructure using 32 TPUv4 chips in MegaCore mode, by using the Pathways orchestration layer (Barham et al., 2022).

Table 7 reports more details for each training segment.

Table 7: Details for the different training segments of the large scale continual learning experiment described in Section 5.

| Training | TPU | | | #Tasks | #Params | Activated params |
|---|---|---|---|---|---|---|
| segment | core-hours | #cores | type | | (M) | per task (%) |
| ViT tasks 10 iters | 4949 | 8 | TPUv3 | 3 | 659 | 46.6% |
| VTAB-full 2 iters | 5927 | 32 | TPUv4 | 26 | 4400 | 7.0% |
| ViT tasks +1 iter | 541 | 32 | TPUv4 | 26 | 4577 | 5.2% |
| VDD 2 iters | 1507 | 32 | TPUv4 | 36 | 7790 | 3.9% |
| ViT tasks +1 iter | 564 | 32 | TPUv4 | 36 | 7854 | 3.9% |
| VTAB-full +1 iter | 2266 | 32 | TPUv4 | 36 | 7596 | 4.0% |
| Char. class. 2 iters | 1785 | 32 | TPUv4 | 44 | 9368 | 3.3% |
| VTAB-1k 2 iters | 271 | 32 | TPUv4 | 69 | 13087 | 2.3% |
| ViT tasks +1 iter | 568 | 32 | TPUv4 | 69 | 13090 | 2.3% |

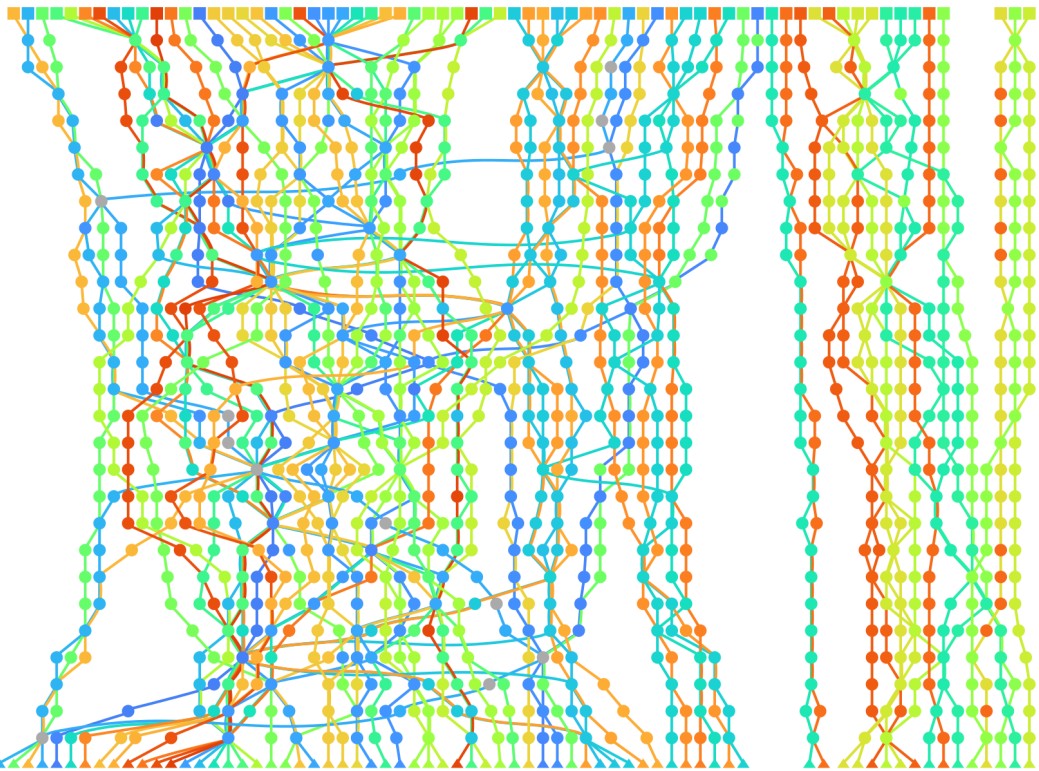

Figure 3: Graph representing the architecture of the multitask system solving jointly 69 image classification tasks generated by the large scale continual learning experiment described in Section 5. Each task is identified with a unique color. Bottom triangular nodes represent the data input of each task. Top rectangular nodes represent the head layer of each task. Each edges sequence of the same color connecting a task input to its head, a *path*, defines the layers sequence composing the model for each task. Each path traverses 27 round nodes representing ViT L/16 internal layers (see Section 3.5) in the following order from bottom to top: patch embedding, class token, position embedding and 24 transformer layers. Internal nodes are represented with the color of the task on which the parameters of the corresponding layer were trained last. Except for the gray nodes that have not received gradient updates from any of the 69 tasks and still carry the parameters of the root model that were loaded from a checkpoint of a ViT L/16 pretrained on the imagenet-21k dataset (see Section 5) (Video: youtu.be/Hf88Ge0eiQ8).

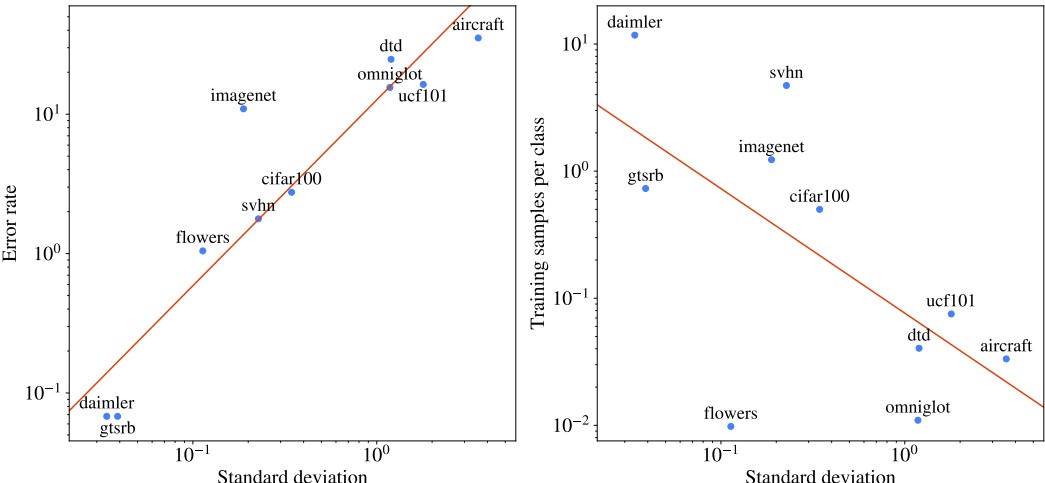

Figure 4: Correlation in log scale with the standard deviation measured during the variance analysis conducted on Visual Domain Decathlon benchmark by running the training on 3 parallel replicas of the system. We display the 2 metrics that are most correlated with the standard deviation: error rate computed on the test set (left) and training samples per class (right). The red line is fitted to minimize the squared distance to the set of points.

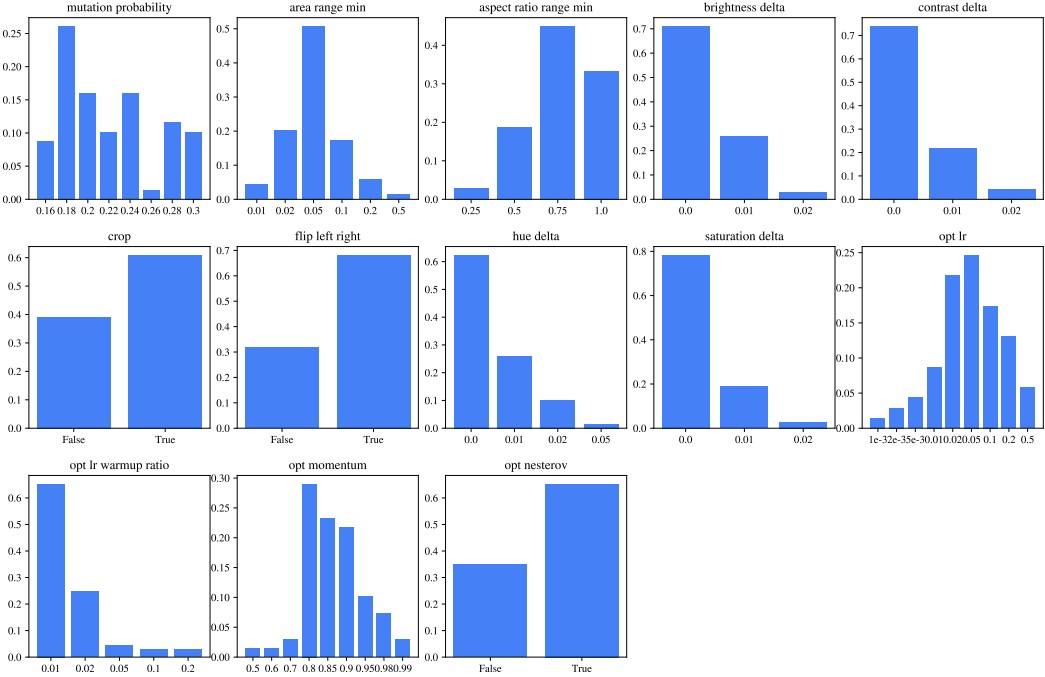

Figure 5: Distributions of the hyperparameter values used by the best models of the 69 image classification tasks at the end of the large scale continual learning experiment described in Section 5.

Table 8: Datasets splits and reference (part 1 of 2). For each dataset used in the experiments, this table reports: 1) dataset name indicative of the Tensorflow Datasets Catalogs identification string and linking to the corresponding catalog page ("visual_domain_decathlon" has been abbreviated as "vdd", and "diabetic_retinopathy_detection" as "drd"), 2) train, validation and test data splits, represented with the standard Tensorflow Datasets format ("validation" has been abbreviated as "val"). 3) corresponding scientific publication reference. Datasets are listed in the order of introduction into the system.

Notes:

[1] The test split of the imagenet_v2 dataset is used as validation set for imagenet2012.

[2] The test split of the cifar10_1 dataset is used as validation set for cifar10.

[3] The VTAB-full benchmark also includes the cifar100 task. Cifar100 has been introduced to the $\mu$2Net system as part of the initial benchmark. In the VTAB-full results tables we refer to the top 1 test accuracy achieved in the latest cifar100 training iteration without retraining it as part of the VTAB-full active training iteration.

[4] The definition for the VTAB standard and additional tasks has been sourced from https://github.com/google-research/task_adaptation/tree/master/task_adaptation/data.

[5] VTAB additional task, not included in the standard scoring set. These tasks were added to further scale the system and analyze transfer across related tasks.

| Name | Splits | | | Reference |
| --- | --- | --- | --- | --- |
| | Train | Val. | Test | |
| imagenet2012 | train | imagenet_v2:test[1] | val | (Russakovsky et al., 2015) |
| cifar100 | train[:98%] | train[98%:] | test | (Krizhevsky, 2009) |
| cifar10 | train | cifar10_1:test[2] | test | (Krizhevsky, 2009) |
| **VTAB-full benchmark**[3][4] | | | | |
| caltech101 | train[:2754] | train[2754:] | test | (Fei-Fei et al., 2004) |
| dtd | train | val | test | (Cimpoi et al., 2014) |
| oxford_flowers102 | train | val | test | (Nilsback & Zisserman, 2008) |
| oxford_iiit_pet | train[:2944] | train[2944:] | test | (Parkhi et al., 2012) |
| sun397 | train | val | test | (Xiao et al., 2010) |
| svhn_cropped | train[:65931] | train[65931:] | test | (Netzer et al., 2011) |
| patch_camelyon | train | val | test | (Veeling et al., 2018) |
| eurosat/rgb | train[:16200] | train[16200:21600] | train[21600:] | (Helber et al., 2019) |
| resisc45 | train[:18900] | train[18900:25200] | train[25200:] | (Cheng et al., 2017) |
| drd/btgraham-300 | train | val | test | (Kaggle & EyePacs, 2015) |
| clevr/count_cylinders[5] | train[:63000] | train[63000:] | val | (Johnson et al., 2017) |
| clevr/count_all | train[:63000] | train[63000:] | val | (Johnson et al., 2017) |
| clevr/closest_object_distance | train[:63000] | train[63000:] | val | (Johnson et al., 2017) |
| dmlab | train | val | test | (Zhai et al., 2019a) |
| dsprites/label_x_position | train[:589824] | train[589824:663552] | train[663552:] | (Klindt et al., 2021) |
| dsprites/label_orientation | train[:589824] | train[589824:663552] | train[663552:] | (Klindt et al., 2021) |
| kitti/closest_object_distance[5] | train | val | test | (Geiger et al., 2012) |
| kitti/count_vehicles[5] | train | val | test | (Geiger et al., 2012) |
| kitti/closest_vehicle_distance | train | val | test | (Geiger et al., 2012) |
| smallnorb/label_category[5] | train | test[:50%] | test[50%:] | (LeCun et al., 2004) |
| smallnorb/label_lighting[5] | train | test[:50%] | test[50%:] | (LeCun et al., 2004) |
| smallnorb/label_azimuth | train | test[:50%] | test[50%:] | (LeCun et al., 2004) |
| smallnorb/label_elevation | train | test[:50%] | test[50%:] | (LeCun et al., 2004) |
| **Visual domain decathlon benchmark** | | | | |
| vdd/imagenet12 | train | val[:50%] | val[50%:] | (Bilen et al., 2017) |
| vdd/svhn | train | val[:50%] | val[50%:] | (Bilen et al., 2017) |
| vdd/cifar100 | train | val[:50%] | val[50%:] | (Bilen et al., 2017) |
| vdd/gtsrb | train | val[:50%] | val[50%:] | (Bilen et al., 2017) |
| vdd/daimlerpedcls | train | val[:50%] | val[50%:] | (Bilen et al., 2017) |
| vdd/omniglot | train | val[:50%] | val[50%:] | (Bilen et al., 2017) |
| vdd/ucf101 | train | val[:50%] | val[50%:] | (Bilen et al., 2017) |
| vdd/aircraft | train | val[:50%] | val[50%:] | (Bilen et al., 2017) |
| vdd/dtd | train | val[:50%] | val[50%:] | (Bilen et al., 2017) |
| vdd/vgg-flowers | train | val[:50%] | val[50%:] | (Bilen et al., 2017) |

Table 9: Datasets splits and reference (part 2 of 2).

| Name | Splits | | | Reference |
|---|---|---|---|---|
| | Train | Val. | Test | |
| ...Continues from Table 8 | | | | |
| **Multitask Character Classification Benchmark** | | | | |
| emnist/digits | train[5%:] | train[:5%] | test | (Cohen et al., 2017) |
| emnist/letters | train[5%:] | train[:5%] | test | (Cohen et al., 2017) |
| kmnist | train[5%:] | train[:5%] | test | (Clanuwat et al., 2018) |
| mnist | train[5%:] | train[:5%] | test | (LeCun et al., 1998) |
| omniglot | train | small1 | small2 | (Lake et al., 2015) |
| cmaterdb/bangla | train[20%:] | train[:20%] | test | (Das et al., 2012b;a) |
| cmaterdb/devanagari | train[20%:] | train[:20%] | test | (Das et al., 2012b;a) |
| cmaterdb/telugu | train[20%:] | train[:20%] | test | (Das et al., 2012b;a) |
| **VTAB 1k benchmark**[4] | | | | |
| caltech101 | train[:800] | train[2754:2954] | test | (Fei-Fei et al., 2004) |
| cifar100 | train[:800] | train[45000:45200] | test | (Krizhevsky, 2009) |
| cifar10 | train[:800] | train[45000:45200] | test | (Krizhevsky, 2009) |
| dtd | train[:800] | val[:200] | test | (Cimpoi et al., 2014) |
| oxford_flowers102 | train[:800] | val[:200] | test | (Nilsback & Zisserman, 2008) |
| oxford_iiit_pet | train[:800] | train[2944:3144] | test | (Parkhi et al., 2012) |
| sun397 | train[:800] | val[:200] | test | (Xiao et al., 2010) |
| svhn_cropped | train[:800] | train[65931:66131] | test | (Netzer et al., 2011) |
| patch_camelyon | train[:800] | val[:200] | test | (Veeling et al., 2018) |
| eurosat/rgb | train[:800] | train[16200:16400] | train[21600:] | (Helber et al., 2019) |
| resisc45 | train[:800] | train[18900:19100] | train[25200:] | (Cheng et al., 2017) |
| drd/btgraham-300 | train[:800] | val[:200] | test | (Kaggle & EyePacs, 2015) |
| clevr/count_cylinders[5] | train[:800] | train[63000:63200] | val | (Johnson et al., 2017) |
| clevr/count_all | train[:800] | train[63000:63200] | val | (Johnson et al., 2017) |
| clevr/closest_object_distance | train[:800] | train[63000:63200] | val | (Johnson et al., 2017) |
| dmlab | train[:800] | val[:200] | test | (Zhai et al., 2019a) |
| dsprites/label_x_position | train[:800] | train[589824:590024] | train[663552:] | (Klindt et al., 2021) |
| dsprites/label_orientation | train[:800] | train[589824:590024] | train[663552:] | (Klindt et al., 2021) |
| kitti/closest_object_distance[5] | train[:800] | val[:200] | test | (Geiger et al., 2012) |
| kitti/count_vehicles[5] | train[:800] | val[:200] | test | (Geiger et al., 2012) |
| kitti/closest_vehicle_distance | train[:800] | val[:200] | test | (Geiger et al., 2012) |
| smallnorb/label_category[5] | train[:800] | test[:200] | test[50%:] | (LeCun et al., 2004) |
| smallnorb/label_lighting[5] | train[:800] | test[:200] | test[50%:] | (LeCun et al., 2004) |
| smallnorb/label_azimuth | train[:800] | test[:200] | test[50%:] | (LeCun et al., 2004) |
| smallnorb/label_elevation | train[:800] | test[:200] | test[50%:] | (LeCun et al., 2004) |

---

**Algorithm 1** Pseudocode for one active task iteration

---

1: Active task: $t$
2: Set of all the models currently in the multitask system: $\mathcal{M}$
3: Active population: $\mathcal{A} \leftarrow \{m \mid m \in \mathcal{M} \land m \text{ trained on } t\}$
4: **for** $\#generations$ **do**
5:     **for** $\#child\text{-}models$ **do**
6:                                                                   ▷ Sample parent model
7:         Parent model: $p \leftarrow$ **none**
8:         **for** Candidate parent model: $\hat{p} \in [sorted_{score}(\mathcal{A}), sorted_{random}(\mathcal{M} \setminus \mathcal{A})]$ **do**
9:             **if** $0.5^{\#selections(\hat{p},t)} > x \sim Uniform([0,1])$ **then**
10:                 $p \leftarrow \hat{p}$
11:                 **break**
12:             **end if**
13:         **end for**
14:         **if** $p =$ **none then**
15:             $p \sim Uniform(\mathcal{A} \cup \mathcal{M})$
16:         **end if**
17:                                                               ▷ Sample child model
18:         Set of mutations: $\Delta \leftarrow \{make\text{-}trainable\text{-}head\}$
19:         **for** Candidate mutation: $\hat{\delta} \in possible\text{-}mutations(p)$ **do**
20:             **if** $\mu > x \sim Uniform([0,1])$ **then**
21:                 $\Delta \leftarrow \Delta \cup \{\hat{\delta}\}$
22:             **end if**
23:         **end for**
24:         Untrained child model: $c_0 \leftarrow apply\text{-}mutations(p, \Delta)$
25:                                                                 ▷ Train child model
26:         Retained child model: $c \leftarrow$ **none**
27:         **for** $i \in [1, \dots, \#train\text{-}cycles]$ **do**
28:             $c_i \leftarrow train(c_{i-1}, min(1\ epoch,\ \#samples\text{-}cap))$
29:             **if** $score(c_i) \geq max(\{score(c) | c \neq \textbf{none}\} \cup \{score(p) | p \text{ trained on } t\} \cup \{-\infty\})$ **then**
30:                 $c \leftarrow c^i$
31:             **end if**
32:         **end for**
33:         **if** $c \neq$ **none then**
34:             $\mathcal{A} \leftarrow \mathcal{A} \cup \{c\}$
35:         **end if**
36:     **end for**
37: **end for**
38:                                                                 ▷ Keep only the best model for $t$
39: $\mathcal{M} \leftarrow \{\text{argmax}_{m \in \mathcal{A}}\ score(m)\} \cup \{m \mid m \in \mathcal{M} \land m \text{ not trained on } t\}$

---

