# OpenReview forum: "An Evolutionary Approach to Dynamic Introduction of Tasks in Large-scale Multitask Learning Systems"
_ICLR.cc/2023/Conference — Submitted to ICLR 2023_

### Official Review · Reviewer_uLXC · 2022-10-24

**Confidence:** 3
**Correctness:** 2
**Technical Novelty And Significance:** 1
**Empirical Novelty And Significance:** 1
**Recommendation:** 1

**Clarity, Quality, Novelty And Reproducibility:**

This paper should be thoroughly reorganized. Its novelty and reproducibility are not yet satisfactory for high-quality magazines.

**Strength And Weaknesses:**

Strength
1.	This paper aims to provide a new and automatic method for dynamic large-scale multitask learning.

Weaknesses
1.	The main contribution of this paper is unclear. Although it claimed that the proposed method possesses 8 novel properties, they either somewhat overstated the ability or applicability of the proposed method or were not well-supported. The main idea of how the proposed method copes with dynamic large-scale multitasking is not clear. How the automation is achieved is also unclear.
2.	A more comprehensive review should be provided. How existing works deal with dynamic multitasking problems and what is the scale of the problems they face are not well presented.
3.	The problem studied in this paper is not well-formulated. It is not clear what is meant by "large-scale" as well as how large is large in this paper.


**Summary Of The Paper:**

This paper proposes a new method for dynamic large-scale multitask learning and conducts some experiments to study its effectiveness.

**Summary Of The Review:**

This paper studies dynamic large-scale multitask learning problems. However, the problem is not well-formulated and the proposed method is not clearly described. It is poorly organized.

---

### Official Review · Reviewer_uZHy · 2022-10-24

**Confidence:** 4
**Correctness:** 3
**Technical Novelty And Significance:** 2
**Empirical Novelty And Significance:** 3
**Recommendation:** 6

**Clarity, Quality, Novelty And Reproducibility:**

The paper proposes a novel evolutionary method for multi-task architecture learning. It is clearly written and reproducible by the submitted code.

**Strength And Weaknesses:**

Strength

- The method improves the baselines fine-tuned on individual tasks on some datasets (ViT and VTAB benchmarks).
- One strength is its extensive experiments on various datasets, including the large-scale evaluation on 69 classification tasks.
- The training procedure is visualized as a video to show the evolution of the model, which is very helpful in understanding the algorithm. The paper also provides various interesting analyses of the proposed algorithm.
- The paper proposes a novel method that sequentially extends the model to enhance the accuracy of individual tasks by transferring knowledge over tasks.
- The code is submitted, and the results will be reproducible.

Weakness

- I am not sure about the effectiveness of the search algorithm because the search space consisting of all combinations of mutations is very large, but the method searches for a few iterations with several generations and childs. Ideally, different trials should converge to a similar solution regardless of the task orders used during training. Will this be the case? I think it would be nice to include such a discussion.
- Is there a specific reason to use only 2 active task iterations? Will the longer iterations help find better architecture and further improve the accuracy, or is it already saturated with 2 iterations?

- Computational cost looks pretty high, because it requires to train models for #task * #(active task iteration) * #generation * #child times.

- It would be nice if the effect of the order of active tasks can be studied. The method can be used for multi-task learning, as well as continual learning, which allows access to the previous task data during training. For the multi-task learning setting, what will be the optimal root model to start from? Also, what will be the best order of task activations?

- Experimental validation seems not sufficient. There are existing methods that also allow transferring knowledge across tasks, such as PNN [A] and DEN [B], by extending architectures for a sequence of tasks. Comparison with such methods in the experiments will be necessary. In addition, I think a multi-task model with a shared backbone (e.g. multi-head model, classification model for the unified label space) is another baseline to compare with, which could be a simple but effective way to allow knowledge transfer across-task.

- Current experiments are all restricted to classification tasks, which are relatively similar tasks. It would be interesting to see if the method applies to more general tasks beyond classification tasks. For example, can this method apply to dense prediction tasks, such as segmentation, depth estimation, and surface normal estimation (Taskonomy, NYUv2, Pascal-Context), where there can be clear conflict across tasks?

Other minor things
- Analysis of some hyper-parameters is missing. How is the search space decided? What is the effect of different search spaces (different ranges of hyper-parameters)? How is the number of cycles for validation decided, and what is the effect of different choices? Does it need to be long enough to ensure the model is converged? How do you select the training and validation set to use?
- How is the order of active tasks decided? What is the effect of different orders and choices of the root task model?
- Is Zahi et al in Table 3 share the same backbone with the proposed model, being comparable?
- The purpose of Figure 4 is unclear to me. Is it to claim that the method improves more if the task is more difficult (higher variance)? Also, in Figure 4, training samples per class are less than 1 for some tasks. Does it mean there can be 0 sample for some classes?
- The purpose of section 5.5 VTAB-1k benchmark (Table 6) is unclear to me.


[A] Progressive Neural Networks, arXiv 2016
[B] Lifelong Learning with Dynamically Expandable Network, ICLR 2018

**Summary Of The Paper:**

The key idea is to share common modules among tasks to improve accuracy through cross-task knowledge transfer. Specifically, the paper proposes an evolutionary algorithm that iteratively trains the model over tasks where its sub-network for each task is evolved sequentially by mutating subsets of modules or hyper-parameters. Experiments show that the method improves over iterations on several datasets, and outperforms the single-task fine-tuning baseline on ViT and VTAB benchmarks.

**Summary Of The Review:**

In summary, it was an interesting paper with a novel method and extensive experiments. I think additional comparisons with existing works will be necessary.

---

### Official Review · Reviewer_3FfT · 2022-10-25

**Confidence:** 4
**Correctness:** 4
**Technical Novelty And Significance:** 4
**Empirical Novelty And Significance:** 4
**Recommendation:** 6

**Clarity, Quality, Novelty And Reproducibility:**

Clarity: clear enough
quality: experiments are extensive and convincing
Novelty: definitely novel
reproducibility: result are absolutely not reproducible if you have not a huge computational power.

**Strength And Weaknesses:**

strenghts

    - The evolutionary algorithm proposed is quite simple, straightforward and seems to work very well both in terms of performance and knowledge sharing. The idea of always freezing the parent models and only allowing weight updates through novel generations allows the method to be immune to typical continual learning issues such as catastrophic forgetting and typical multi-task issues such as gradient interference.
    - The experimental section of the paper covers an extremly broad range of tasks and validates several claims made in the paper. Overall the section is very extensive.
    - The results on all the benchmarks are quite impressive.


weaknesses

    - Fig. 1 leaves me with some doubts. It would seem that the private task is solved by using only a head operating on the learned layer for the green task (devanagari). This is at least what I would expect for the claims of the method to still uphold, because if the private task head can alter the weights of the Transformer layer 1 then information from the private task is flowing into the network. I would appreaciate if the authors could clarify this.
    - Overall a lot of choices seem to lead towards the necessity for large compute power. The choice of modifying hyperparameters only by a one-hop neighbor is quite restrictive and it implies that we have to evolve/search for quite a while before stumbling on the correct hyperparams. The layer cloning and mutation probability hyperparameter is set at random by the evolutionary process, implying that the level of overall randomicity is very high and therefore large training times are needed to get stable results or be able to reproduce the claimed results (considering the authors use DNN architectures). The authors mention that "the score can be defined to optimize a mixture of factors depending on application requirements". It would have been nice to see what the tradeoff between training time and model size vs optimal multi-task performance is, especially considering these high levels of randomicity present in the proposed approach. (and also for others to be able to reproduce somewhat similar results on lesser compute).
    - The parameters in Table 1, the model and the experiments seem to be only good for image data and ViT. Did the authors try to apply the same principles to other areas research areas such as NLP or simpler models in the image domain (CNNs)? I understand the latter might be due to the focus about state of the art performance, but it would show that the method can generalize to different architectures and tasks, not just transformers in vision.


**Summary Of The Paper:**

The authors present an evolutionary algorithm capable of dynamically building a large and performing multi-task model over time. The prosed method can be considered a union of multi-task learning and continual learning, while trying to relate both to the more general concept of knowledge trasfer through experience. The authors present the algorithm and a large set of experiments to validate their claims

**Summary Of The Review:**

Great results given a huge computational power.

---

### Official Review · Reviewer_W895 · 2022-10-31

**Confidence:** 4
**Correctness:** 2
**Technical Novelty And Significance:** 2
**Empirical Novelty And Significance:** 2
**Recommendation:** 5

**Clarity, Quality, Novelty And Reproducibility:**

Clarity. The paper is very clear.
Quality. The paper is well written. I appreciate the supplementary visual and video material.
Novelty. This works lacks novelty. I found it hard to identify the novel elements of the proposed evolutionary approach.
Reproducibility. Sufficient details provided to reproduce experiments and results.

**Strength And Weaknesses:**

Strengths
+ The paper focuses on large scale datasets and tasks which is challenging in multi-task learning and continual learning.
+ Comprehensive Experiments

Weaknesses
- No baselines (except ViT [Dosovitskiy et al., 2021], [Steiner et al., 2021], and [Zhai et al., 2019b] in some but not all experiments) despite the presence of recent similar methods such as RCL [Xu and Zhu, 2018] and DEN [Yoon, 2018] as well as of course well-known paradigms such as multi-task learning.
- "The presented experiments aim to compare against the state of the art for a large number of tasks without any size or compute constraint. Therefore, the validation accuracy is used directly as the score without additional factors." This is a major weakness of the paper especially if the focus is on large-scale multi-task learning systems.
- The empirical evaluation lacks any mention or analysis of model complexity (especially compared with baselines).

**Summary Of The Paper:**

This paper proposes an evolutionary approach, $\mu$2Net, to dynamically generate a large scale multitask network for multiple tasks. The evolutionary approach comprises (1) sampling a parent model, (2) mutations, (3) training and scoring children models. Extensive experiments are performed on numerous datasets.

**Summary Of The Review:**

Interesting paper but lacks sufficient novelty and comprehensive empirical evaluation.

---

### Decision · Program_Chairs · 2023-01-20

**Decision:**

Reject

**Justification For Why Not Higher Score:**

The authors did not answer any questions raised by the reviewers.

**Justification For Why Not Lower Score:**

N/A

**Metareview: Summary, Strengths And Weaknesses:**

This paper proposes a new evolutionary approach to designing large-scale multi-task models. While the reviewers raised several concerns about the paper, the authors did not make any effort in writing a rebuttal to address these concerns. Hence I recommend rejection.